# Evolution of Volatile Compounds during Ripening and Final Sensory Changes of Traditional Raw Ewe’s Milk Cheese “Torta del Casar” Maturated with Selected Protective Lactic Acid Bacteria

**DOI:** 10.3390/foods11172658

**Published:** 2022-09-01

**Authors:** Irene Martín, Alicia Rodríguez, Carmen García, Juan J. Córdoba

**Affiliations:** 1Higiene y Seguridad Alimentaria, Instituto Universitario de Investigación de Carne y Productos Cárnicos (IProCar), Facultad de Veterinaria, Universidad de Extremadura, Avda. de las Ciencias, s/n., 10003 Cáceres, Spain; 2Tecnología y Calidad de Alimentos, Instituto Universitario de Investigación de Carne y Productos Cárnicos (IProCar), Facultad de Veterinaria, Universidad de Extremadura, Avda. de las Ciencias, s/n., 10003 Cáceres, Spain

**Keywords:** soft ripened cheese, lactic acid bacteria, protective cultures, volatile compounds, texture, sensorial analysis

## Abstract

In traditional soft ripened cheeses made with raw milk, the use of protective cultures is infrequent. In the present work, the effect of selected (for their activity against *Listeria monocytogenes*) protective cultures of *Lactocaseibacillus casei* 116 and *Lactococcus garvieae* 151 was evaluated, on the evolution of volatile compounds throughout the ripening and on the final sensory characteristics of traditional soft ripened “Torta del Casar” cheese. For this, both strains were separately inoculated in raw cheeses and ripened for 90 days. The selected LAB strains did not affect physicochemical parameters, including texture and color of the ripened cheeses. However, they could have a positive effect on the aroma, for the generation of methyl branched acids and for the reduction in compounds derived from β-oxidation of fatty acids. Thus, these protective cultures, in addition to contributing to their safety, could improve quality of the ripened cheeses.

## 1. Introduction

“Torta del Casar” is a high-quality Spanish cheese marketed under the Registry of the Protected Designation of Origin (PDO) “Torta del Casar” (Casar de Cáceres, Cáceres, Spain) in accordance with Regulation (CE) 1491/2003 [1] of the European Commission. This type of cheese is made from whole raw Merino ewe’s milk using only the dried flowers of the plant *Cynara cardunculus* as rennet. In this kind of cheese, the content of protein and lipid is around 21% and 29%, respectively [2]. The absence of any standardizing thermal process leads to the presence of undesirable microorganisms throughout ripening [3]. In recent decades soft cheeses such as “Torta del Casar” have been linked to many outbreaks of illnesses in Europe and worldwide [4,5]. Due to its ubiquity and ability to survive and even grow at refrigeration temperatures, most of the outbreaks related to the consumption of soft cheeses have been caused by *Listeria monocytogenes* [6,7]. 

The use of bioprotective cultures represents an additional hurdle to avoid *L. monocytogenes* proliferation and persistence in these products [5,8]. Lactic-acid bacteria (LAB) are the most frequent microbial population of raw milk [9] and can be used as protective cultures in this product. However, to date, the use of protective cultures is not regulated in the PDO “Torta del Casar” cheese [1] and it has not been reported in the literature either.

*Lacticaseibacillus casei* 116 and *Lactococcus garvieae* 151 strains have been previously selected by their antagonistic effect against *L. monocytogenes* in “Torta del Casar” cheese [10]. *L. casei* isolated from dairy products has been associated to antimicrobial activity, improvement of sensory characteristic of milk products and even functional activities as probiotic [11]. *Lc. garvieae* has been reported in dairy products such as goat cheese and raw cow milk [12]. In addition, strains of this species isolated from dairy products has been proposed as a probiotic for controlling the pathogenic *Staphylococcus aureus* in cheese made with raw milk [13]. However, before proposing some of these strains as possible protective cultures in this kind of cheese, it is necessary to evaluate that there are no negative effects on volatile compound generation throughout ripening and in the final sensorial characteristics of this product. 

The aim of the present work was to evaluate the effect of the addition of selected LAB strains isolated from raw ewe´s milk cheeses on the evolution of the volatile compounds throughout the ripening process and the final sensory traits of “Torta del Casar” cheese.

## 2. Materials and Methods

### 2.1. Origin of the Strains and Growth Conditions

The *L. casei* 116 (No. P202131120) and *Lc. garvieae* 151 strains maintained at the Food Hygiene and Safety Culture Collection at the University of Extremadura (Cáceres, Spain), have been used for the inoculation of traditional raw ewe´s milk cheese. These strains were isolated from traditional ripened soft cheeses and selected by its antagonist activity against *L. monocytogenes* in cheese-based agar following the methodology described by Martín et al. [14]. 

To prepare LAB inoculum, 100 μL of the stock culture (stored in Man Rogosa Sharpe (MRS) broth (Fisher Bioreagents, Madrid, Spain) containing 20% (*w*/*v*) glycerol at −80 °C) were inoculated onto 10 mL of MRS broth and incubated for 48 h at 30 °C. At the end of the incubation, ≈8.0 log CFU/mL cells were obtained and an aliquot of this was diluted in 1% (*w*/*v*) peptone water (Conda, Madrid, Spain) to reach a final concentration of approximately 7.0 log CFU/mL. Then, cultures were centrifuged at 10,000× *g* for 5 min, and the supernatants were discarded. The sediments were then washed and resuspended in phosphate-buffered saline (PBS, Fisher Bioreagents) and used for the inoculation of the “Torta del Casar” cheese after salting. To verify the level of inoculation, serial dilutions were poured onto MRS agar (Oxoid, Basingstoke, UK) and incubated anaerobically at 30 °C for 72 h. In addition, the initial counts (CFU/g) of *L. casei* 116 and *Lc. garvieae* 151 on the curd were determined at day 0 of processing.

### 2.2. Preparation of “Torta del Casar” Cheese

To avoid contamination in the industry with the two assayed LAB strains (*L. casei* 116 and *Lc. garvieae* 151), since their industrial use are not allowed at the moment, the “Torta del Casar” cheeses used in this study were first elaborated in a cheese factory located in Cáceres (Extremadura region, Spain) and immediately, in the first day of processing, were transported to the pilot plant of the Faculty of Veterinary under refrigerated conditions where they were inoculated. After that, the cheeses were ripened for 90 days. In this factory, the curds from three different cheeses were made, using raw milk from at least five different farms, with an approximate weight of 0.5 kg. They were elaborated by pressing for 1.5 h and salted in brine for 1 h. After salting, the curds were transported from the factory to the Faculty of Veterinary at <2 °C for their inoculation. The inoculation was carried out in the center of the curd (in a cube with a 16 cm^2^ surface and a depth of 6 cm [the entire volume of curd], and ≈100 g of weight). According to the inoculated microorganisms, three batches of cheeses were manufactured: batch C (uninoculated control), batch Lc (inoculated only with *L. casei* at ≈7 log CFU/g) and batch Lg (inoculated only with *Lc. garvieae* ≈7 log CFU/g). The two bacterium inocula were prepared each in a final volume of 1 mL of PBS with sterile micropipettes in a laminar flow cabinet (Telstar, Spain). In batch C, 1 mL of sterilized PBS was added instead of the bacterium inoculum. 

After inoculation, the cheeses’ curds were ripened in a chamber of the pilot plant following the industrial conditions used for this product: 35 days at 6 °C and 90% relative humidity (RH), 10 days at 8 °C and 80 % RH, 10 days at 9 °C and 80 % RH. Finally, cheeses were kept at 10 °C and 80% RH for 35 days.

Five cheeses of each batch were taken at 0, 30, 45, 60 and 90 days of ripening for microbiological, physicochemical, and volatile compound analysis. In ripened cheeses, color, texture, and sensory analysis were also determined. All analyses were carried out in quintuplicate. For the microbiological analysis, the entire cube (16 cm^2^ of surface and 6 cm deep) of cheeses was used during all sampling times. Thus, the experiment consisting of 3 different batches ×5 sampling times ×5 different cheese/each batch and sampling time, which were evaluated once, according to the European Union Reference Laboratory Technical Guidance Document for conducting shelf-life studies on *L. monocytogenes* in RTE foods (such as “Torta del Casar” cheese) where no growth or the growth probability of this pathogen is ≤10% [15].

### 2.3. Microbiological Analysis

The total viable aerobic microbial and LAB counts were determined on Plate Count Agar (PCA; Pronadisa, Spain) and MRS (Oxoid, Basingstoke, UK) agar, respectively. Both agar media were incubated at 30 °C for 48 h under aerobic and microaerophilic conditions, respectively. Finally, the *Enterobacteriaceae* were counted on Violet Red Bile Glucose (VRBG, Oxoid, Basingstoke, UK) agar, and the incubation was carried out at 37 °C for 48 h under aerobic conditions. After incubation, colonies with the expected characteristics were counted and results were expressed as log CFU/g.

The evaluation of the implantation of *L. casei* 116 and *Lc. garvieae* 151 in the batches Lc and Lg was conducted in MRS plates in the last sampling time (90 days) following the procedure described by Martín et al. [16]. The identification of the LAB strains was performed by sequencing analysis of the 16S rRNA region according to the methodology proposed by Walter et al., [17] and PFGE analysis of the DNA following procedures previously described by Alía et al. [18]. Most of the investigated LAB isolates of Lc and Lg were identified as *L. casei* 116 and *Lc. garvieae* 151, respectively, by sequencing analysis of the 16S rRNA region and PFGE analysis, confirming the implantation of these strains in the inoculated batches. 

### 2.4. Physicochemical Analysis

The water activity (a_w_) of “Torta del Casar” cheeses was determined at 25 °C by using a Novasina Lab Master meter (Novasina AG, Lachen, Switzerland). Calibration was achieved by using several saturated solutions of known a_w_. The pH was measured using a pH-meter model 340 (Mettler-Toledo GmbH, Greifensee, Switzerland) calibrated with 3 different standard pH solutions (4.0, 7.0 and 9.25). Moisture content (%) was gravimetrically analyzed following the official method of the Association of Official Analytical Chemists [19]. 

### 2.5. Instrumental Texture

The texture analysis was performed at room temperature using a Texture Profile Analysis (TPA) and was carried out in triplicate of each of the five cheese samples composing each batch at the end or the ripening time. Cutting slices of cheese approximately 1 cm thick was measured. The instrument used was a TA XT Plus Texture Analyzer (StableMicro Systems Ltd., Godalming, UK) equipped with a cylindrical probe of 5 cm in diameter. The TPA settings were as follows: 0.83 mm/s pre- and test speed, 1.67 mm/s post-test speed, deformation of 35% for 0.0833 min and activation force of 0.049 N. In these conditions hardness (N), springiness (cm), cohesiveness, gumminess (N), chewiness (N cm), and adhesiveness (N s) were evaluated. 

### 2.6. Instrumental Color

Color was determined on the cut surface of each sample using a Minolta CR-300 colorimeter (Konica Minolta, Inc; Nieuwegein, The Netherlands) with an illuminant D65, a 0° standard observer and one port/display area of 2.5 cm. that was calibrated before use with a white tile having the following values: L* = 93.5, a* = 1.0 and b* = 0.8. Color was expressed according to the Commission International de l’Eclairage (CIE) system and reported as CIE L* (lightness), CIE a* (redness), CIE b* (yellowness), in which the chroma and hue angle were calculated as (a*^2^ + b*^2^)^0.5^ and tan ^−1^(b*/a*), respectively.

### 2.7. Volatile Compound Analysis

The volatile compounds in soft cheeses were extracted by solid-phase microextraction (SPME) after heating to 37 °C for 30 min, using a divinylbenzene-carboxen-polydimethylsiloxane (DVB/CAR/PDMS) 50/30 µm fiber (Merck; Darmstadt, Germany). They were then analyzed by gas chromatography-mass spectrometry (GC-MS) in a Gas Chromatograph 6890 GC (Agilent Technologies; Santa Clara, CA, USA) equipped with a HP-5 column (5% phenyl−95% dimethylpolysiloxane) and coupled to a mass spectrometer (MS) detector, 5975C (Agilent Technologies). Oven temperature started at 40 °C for 5 min and was increased to 280 °C, with a rate of 7 °C/min. The desorption time was 30 min at 250 °C. The transfer line temperature was established at 280 °C. The carrier gas was helium (Air Liquide, Madrid, Spain) with a flow rate of 1.2 mL/min. MS detection was performed in full scan (50–350 amu). Automated peaks search and spectral deconvolution were used for data treatment, and the identification of the volatile compounds was achieved by comparing their mass spectra with the NIST/EPA/NIH library (Institute of Standards and Technology, Gaithersburg, MD, USA). Volatile compound identifications were confirmed based on comparisons of the linear retention index of standards of a series of n-alkanes analyzed under identical conditions to the samples.

### 2.8. Sensory Evaluation

A triangular of olfactory analysis (discriminant test) was carried out in this study with a semi-trained panel of 24, including students and lectures at the Faculty of Veterinary Sciences (University of Extremadura, Caceres, Spain). The study was conducted in accordance with the Declaration of Helsinki, and approved by the Ethics Committee of the University of Extremadura (reference 77/2017) for studies involving humans. The panel sessions were held around 2 h before lunch in the sensory panel booth room at the Faculty of Veterinary Science of the University of Extremadura in Cáceres (Spain). Information about sex and age of each volunteer was required. Three samples were presented to each volunteer, marked with random three digits’ codes, and served at room temperature on white plastic plates. In this triangular test the null hypothesis establishes that the probability of randomly choosing a different sample is *p* = 1/3 [20] and the level of significance *p* ≤ 0.05. 

### 2.9. Statistical Analyses

For the statistical analysis of the data, the software IBM SPSS Statistic version 20 (IBM, New York, NY, USA) was used. Once the dependent (microbiological level, a_w_, pH, moisture content, texture, color, volatile compounds) and independent (different batches and days of ripening) variables of the analysis were determined, a study of the normality of the different data populations was carried out using the Shapiro–Wilk test. Subsequently, the analysis of the data was conducted using the Mann–Whitney test [21]. Statistical significance was established at *p* ≤ 0.05.

## 3. Results and Discussion

### 3.1. Physicochemical Parameters

The evolution of the physicochemical parameters during ripening is presented in Figure 1. The moisture content (%) decreased significantly (*p* ≤ 0.05) during ripening from initial levels around 88% to 35–38% at the end of ripening in all batches studied (Figure 1a). These values are similar to those found in soft-ripened cheeses by Ordiales et al. [22]. There were no differences between batches C and Lc throughout the ripening period; although, the moisture content values of the batch inoculated with *Lc. garvieae* (Lg) were significantly lower at days 30 and 45 (Figure 1a). However, there were no significant differences (*p* > 0.05) between batches at the end of the ripening time.

The a_w_ ranged from initial values of 0.975 to 0.938 at the end of ripening (Figure 1b). Similar evolution in the a_w_ was reported by Ordiales et al. [22] in “Torta del Casar” cheese. No significant differences (*p* > 0.05) in a_w_ values were found between batches during the ripening. 

The pH values decreased until day 30 of ripening in the three analyzed batches to reach levels of around 5.20 (Figure 1c). However, at days 45, 60 and 90 the pH increased to reach levels around 5.8 at the end of ripening. The increase in pH values of ripened cheese might be due to the decomposition of lactic acid and the formation of basic compounds derived from the hydrolysis of proteins [23,24]. Small differences in the pH values between batches were found, only at day 90, the batch in which *L. casei* was inoculated (batch Lc) showed a pH significantly (*p* ≤ 0.05) higher than that found in the control batch (batch C). 

Thus, only little and punctual differences over the physicochemical parameters of “Torta del Casar” cheese (humidity and pH) as a result of the inoculation of the selected LAB strains *L. casei* and *Lc. garvieae* were observed, but in general, these microorganisms did not affect these parameters on soft-ripened cheeses. These results contrast with those found by Jia et al. [23] in semi-hard goat cheese, who reported that the addition of selected starter cultures could decrease the pH of matured cheese.

### 3.2. Enumeration of Microorganisms

Table 1 shows the results obtained from the enumeration of microorganisms. The total viable aerobic microbial and LAB counts showed similar levels and, higher than 8 log CFU/g, in all batches studied and all sampling days. The total viable microbial and LAB counts were similar to those found in other soft cheeses clotted with vegetable rennet [2,25]. Thus, LAB is the predominant microbial group during the ripening of soft ewe’s milk cheeses made with raw milk and clotted with vegetable rennet, such as “Serra da Estrela”, “Serpa”, “Queso de la Serena” and “Torta del Casar” cheese [26,27].

On the other hand, the levels of *Enterobacteriaceae* were high (≈6.8 log CFU/g) in all batches at the beginning of the ripening (day 0), probably as this kind of cheese is made with raw milk. A significant (*p* ≤ 0.05) decrease in this microbial group was observed throughout the ripening in all batches, including uninoculated control (Table 1), probably due to a_w_ decrease and the antimicrobial effect of the dominant LAB population. Similar levels of *Enterobacteriaceae* have been previously reported in soft ripened cheese made with raw milk [22,28,29]. 

### 3.3. Analysis of Volatile Compounds

A total of 34 volatile compounds were identified in all the three analyzed batches of “Torta del Casar” cheese throughout the 90-day ripening including carboxylic acids, alcohols, aldehydes, ketones and esters (Table 2, Table 3 and Table 4). Most of the identified compounds have also been reported in other studies of “Torta del Casar” cheeses and other types of similar soft-bodied cheeses [22,23,30].

#### 3.3.1. Acids 

Carboxylic acids were the most abundant volatile compounds found (Table 2). The discovery of this wide range of carboxylic acids might be due to the lipolytic LAB activity [23]. Within these compounds, acetic, hexanoic and 3-methyl-butanoic acids were found in higher levels in all batches (Table 2). In the LAB-inoculated batches Lc and Lg, 2-methyl-propanoic acid was also found to be the most abundant one at the end of the ripening period (Table 2). Acetic and hexanoic acids have been reported as the acids found in the highest amounts in ripened “Torta del Casar” cheese [22]. In general, most acids showed significant (*p* ≤ 0.05) increases throughout the ripening process in both the uninoculated control and inoculated batches, especially the methyl-branched acids that were detected at low levels or non-detected at day 0 of ripening (Table 2). The formation of acids in “Torta del Casar” cheese is caused by enzymes, mainly from vegetal rennet, and microbial activity [31,32]. 

Regarding linear n-acids evolution, only scarce differences were found between batches throughout the ripening process. Thus, only acetic acid at day 0, butanoic acid at 45 and 60 and octanoic and propanoic acids at day 30 showed significantly higher amounts in one of the inoculated batches than in the control one (Table 2). In addition, hexanoic acid showed a lower level in batch Lg than in the control batch at 45 days of ripening. However, none of these differences between batches in n-acids abundance was found at the end of ripening, except for propanoic acid that showed a significantly higher amount in batch Lg than in the control batch (Table 2). Acetic and propanoic acids could have a microbial origin as a result of lactose fermentation; mainly some LAB [33]. Short-chain fatty acids have low perception thresholds and provide typical aroma notes to cheeses such as Cheddar, Roncal, Emmental, Camembert and Grana Padano [31]. 

When the evolution of methyl-branched acids is studied, it is remarkable that at day 90 of ripening most of these compounds (2-methyl-propanoic acid, 3-methyl-butanoic acid and 2-methyl-butanoic acid) showed significantly (*p* ≤ 0.05) higher abundance in Lc and Lg than in control batches. Branched-chain carboxylic acids such as 2-methyl-propanoic, 3-methylbutanoic and 2-methylbutanoic acids, derived from the catabolism of valine, leucine and isoleucine, respectively [34], mainly from microbial origin [35]. In the present work the microbial activity of the selected *L. casei* and *Lc. garvieae* strains to generate branched-chain carboxylic acids seems to be demonstrated since they were detected in higher amounts in inoculated rather than control batches. Thus, inoculation of the selected *L. casei* and *Lc. garvieae* could contribute to the aroma of “Torta del Casar” cheese since the 2-methylbutanoic acid and 3-methylbutanoic acids make contributions to the overall aroma and flavor [36]. 

#### 3.3.2. Alcohols, Ketones and Aldehydes

Table 3 shows the results for alcohols, ketones and aldehydes. The alcohols 2-butanol, 2-methyl-1-butanol, 2,3-butanediol, 2-butoxy-ethanol and 2,6-dimethyl-4-heptanol increased their levels throughout the ripening process, while 2-methyl-1-propanol, 3-methyl-1-butanol and phenyl-ethyl-alcohol showed higher abundance at day 0 of ripening and decreased during maturation. The most abundant alcohols at the end of the ripening were 2-butanol and 2,3-butanediol. The 2-butanol derived from 2,3-butanediol by the action of LAB [37], while the microbial reduction in acetoin could be the origin of 2,3-butanediol [38].

Branched-chain alcohols 2-methyl-1-propanol and 3-methyl-1-butanol come from the reduction in branched-chain aldehydes and can be found in raw milk cheeses with intense proteolysis due to vegetable rennet [39] as is the case of “Torta del Casar”. In fact, the former authors found 3-methyl-1-butanol as the major alcohol in “La Serena” cheese also made with vegetable rennet. 

The evolution of the production in alcohol compounds throughout the ripening process was very similar in the three analyzed batches (Table 3). In ripened cheeses the only significant (*p* ≤ 0.05) differences between batches were the lower amount of 2-butanol and 2-methyl-1-propanol in batch Lc than in the control and Lg batches and the lower amount of 2,3-butanediol and 2-butoxy-ethanol in batch Lg than in the remaining batches. 

A low abundance of **ketones** was detected at the beginning of ripening, but significant increases for all detected compounds of this group were found after 90 days of ripening (Table 3). Ketones are abundant constituents of most dairy products and have typical odors and low perception thresholds [23,40]. Thus, they could play an important role in the final aroma of “Torta del Casar” cheese. The 2-butane-one and 2-heptane-one were the most abundant ketones detected during ripening. Similar results have been found in “Torta del Casar” [40] and in other cheeses made with raw sheep’s milk such as “La Serena” cheeses [39]. In the present work, 2-nonanone showed significantly higher abundance in batches Lc and Lg than in control batches, while 2-heptanone and 2-butanone were encountered in higher amounts in the control than in the LAB-inoculated batches (Table 3). These compounds are derived from β-oxidation of fatty acids, that are first oxidized to α-ketoacids, which are further decarboxylated to their corresponding methyl-ketones with one carbon atom less, such as 2-heptanone, 2-nonanone and 2-butanone and finally the methyl-ketones can be reduced to secondary alcohols [41]. These compounds are necessary to cheese aroma, but since they are derived from β-oxidation of fatty acids, their production should be not stimulated, as could happen with the inoculation of selected LAB assayed, to avoid rancidity notes.

As aldehydes, only 3-methylbutanal was identified throughout the ripening process. The aldehydes have been reported as minor compounds in these kind of soft cheeses, probably due to their instability since they are reduced to alcohols or oxidized to acids [40]. High concentrations of aldehydes are associated with the development of off-flavors in cheese [42] so an increase throughout the processing would be negative for the global aroma of this product. In the present work, a significant (*p* ≤ 0.05) increase in 3-methylbutanal was observed throughout the ripening process in all the analyzed batches. The 3-methylbutanal, derived from the degradation of amino acid leucine [43,44], is probably a consequence of proteolysis from vegetal rennet and microbial origin. Nevertheless, since no significant differences (*p* > 0.05) between batches were detected in 3-methylbutanal in the present work, it seems that the selected *L. casei* and *Lc. garvieae* have a low impact on the production of this compound; at least no difference of the probable effect of LAB was present in control samples from contamination origin.

#### 3.3.3. Esters and Other Compounds

A decrease in most of the **esters** throughout the ripening process was observed in all batches (Table 4). Most of the identified esters were ethyl esters, which could be important contributors to the typical aroma of “Torta del Casar” cheeses [30,40]. Some LAB, as well as chemical reactions, are involved in ester formation in soft-ripened cheeses [45]. In the present work, few differences between batches were detected throughout the ripening process, aside from the fact that it is noticeable that in the final product the abundance of hexanoic acid, ethyl ester, octanoic acid, ethyl ester, decanoic acid, ethyl ester and 1-butanol, 3-methyl-, acetate was lower in batch Lc than in control and Lg batches (Table 4). Although some of these esters have a sweet, fruity and ice-cream flavor and could contribute positively to the aroma of ripened cheese, it should be considered that t this also may have a negative influence, depending on the concentration and type of cheese [23]. 

**Other compounds** were also detected throughout the processing in most of the analyzed batches including two pyrazines (trimethyl-pyrazine and 2,5-dimethyl-pyrazine), dimethyl disulfide, and 1,5,9-decatriene, 2,3,5,8-tetramethyl- (Table 4). Only dimethyl disulfide and the alkyl pyrazines increased during the maturation (Table 4). Dimethyl disulfide detected at similar levels in all analyzed batches has been reported in ripened cheeses as degradation of methional that derived from amino acid methionine, but its contribution to the aroma of cheeses is only marginal [46]. Alkyl pyrazines are produced in cheese via the condensation of aminoketones, which are formed mainly through Maillard and Strecker degradation reactions [46]. From the results, only trimethyl-pyrazine showed differences between batches in ripened cheeses. Therefore, batch Lc showed a higher abundance of trimethyl-pyrazine than the remaining batches (Table 4). Trimethyl-pyrazine has been reported in ripened cheeses and it has been associated as being strong savory to musty and potato-like [46]. 

From the analysis of all the above volatile compounds, it can be deduced that the addition to “Torta del Casar” cheese of the selected *L. casei* and *Lc. garvieae* did not negatively affect the flavor development and could even have a positive effect on the aroma, due to their contribution to the generation of methyl branched compounds (mainly methyl branched acids) and as they do not increase and may even reduce oxidation compounds from β-oxidation of fatty acids, as can be observed in Figure 2 where the addition of an abundant area of methyl branched and β-oxidation compounds are represented.

### 3.4. Texture and Colour Analysis

The texture is an important characteristic of cheese in deciding consumer acceptability [40]. The results obtained from the texture and color parameters of the cheese batches at the end of ripening are shown in Table 5. No differences in texture analysis were observed between control (batch C) and LAB-inoculated cheeses (Lc and Lg batches). Medved´ová et al. [47] also did not find significant differences (*p* ≤ 0.05) in the texture of the cheeses when *L. rhamnosus* was added. However, these results contrast with the pronounced effect found by Jia et al. [23] in semi-hard goat cheeses inoculated with selected LAB starter cultures. Regarding color determination, no differences (*p* > 0.05) were observed between those inoculated with *L. casei* (Lc) and control (C) batches in any of the parameters studied. The only differences (*p* ≤ 0.05) observed were those between the inoculated batch with *Lc. garvieae* (Lg) and the control batch in the parameter L* (lightness). The a*(redness) and b*(yellowness) values of all batches were very similar (*p* > 0.05).

### 3.5. Sensory Evaluation

In the triangular olfactory analysis, no differences between batches were found, and no negative effect in the aroma was encountered for any of the panelists in any of the batches analyzed. Thus, the addition of the selected *L. casei* and *Lc. garvieae* to “Torta del Casar” cheese did not provoke detectable changes in the aroma of the product. 

## 4. Conclusions

LAB is the predominant microbial group during the ripening of inoculated and non-inoculated “Torta del Casar” cheeses. The inoculation of selected *L. casei* and *Lc. garvieae* did not affect physicochemical parameters covering humidity, a_w_, pH and texture and color of the ripened cheeses. Carboxylic acids were the most abundant volatile compounds found during ripening and the methyl-branched acids were detected in a higher abundance in inoculated ripened cheeses. No effect in the olfactory evaluation was detected by panelists due to *L. casei* and *Lc. garvieae*, although they could have a positive effect on the aroma, for their contribution to the generation of methyl branched compounds (mainly methyl branched acids) and for not increasing and even reducing oxidation compounds from β-oxidation of fatty acids. 

## 5. Patents

Martín, I., Rodríguez, A and Córdoba, J.J, inventor 2021. New strain of *Lacticaseibacillus casei* 116 with antagonist effect against *Listeria monocytogenes* to be used as protective culture in ripened cheese. No. P202131120.

## Figures and Tables

**Figure 1 foods-11-02658-f001:**
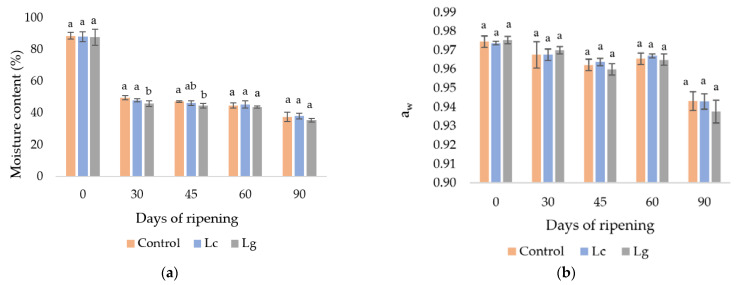
(**a**)Evolution of moisture content; (**b**) water activity and (**c**) pH during ripening of “Torta del Casar” cheese. Mean values with different lowercase letters indicate significant differences between batches at the same incubation day. Control (uninoculated batch), Lc (inoculated with *Lacticaseibacillus casei*) and Lg (inoculated with *Lactococcus garvieae*).

**Figure 2 foods-11-02658-f002:**
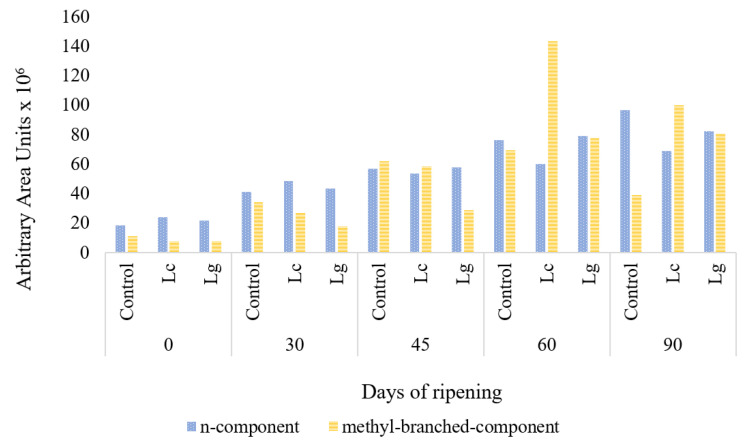
Accumulated area (AU × 10^6^) of n-compounds and methyl branched compounds throughout the ripening of “Torta del Casar” cheese in Control, inoculated with *Lacticaseibacillus casei* (Lc) and inoculated with *Lactococcus garvieae* (Lg) batches.

**Table 1 foods-11-02658-t001:** Counts of total aerobic microorganism (TAM), lactic-acid bacteria (LAB) and *Enterobacteriaceae* (E) throughout “Torta del Casar” cheese ripening.

Batches	Microorganism	Days of Ripening			
		0	45	60	90
C	TAM	8.08 ± 0.034 ^cB^	8.29 ± 0.237 ^aAB^	8.36 ± 0.199 ^bA^	8.37 ± 0.065 ^aA^
	LAB	8.14 ±0.079 ^aA^	8.11 ± 0.216 ^aA^	8.24 ± 0.159 ^aA^	8.32 ± 0.132 ^aA^
	E	6.81 ± 0.294 ^aA^	6.17 ± 0.293 ^aAB^	6.13 ± 0.245 ^aAB^	5.54 ± 0.629 ^aB^
Lc	TAM	8.28 ± 0.046 ^bB^	8.62 ± 0.215 ^aB^	9.06 ± 0.283 ^aA^	8.33 ± 0.141 ^aB^
	LAB	8.30 ± 0.193 ^aA^	8.55 ± 0.158 ^aA^	8.68 ± 0.399 ^aA^	8.14 ± 0.150 ^aA^
	E	6.74 ± 0.307 ^aA^	6.02 ± 0.182 ^aB^	5.81 ± 0.261 ^aB^	5.71 ± 0.323 ^aB^
Lg	TAM	8.41 ± 0.050 ^aAB^	8.5 ± 0.192 ^aA^	8.53 ± 0.355 ^bA^	7.99 ± 0.316 ^bB^
	LAB	8.23 ± 0.057 ^aA^	8.31 ± 0.293 ^aA^	8.50 ± 0.229 ^aA^	7.94 ± 0.314 ^aA^
	E	6.96 ± 0.144 ^aA^	6.13 ± 0.370 ^aB^	5.97 ± 0.097 ^aBC^	5.54 ± 0.255 ^aC^

C (uninoculated batch), Lc (inoculated with *Lacticaseibacillus casei*) and Lg (inoculated with *Lactococcus garvieae*). Values are expressed as mean ± standard deviation. Different lowercase letters indicate significant differences (*p* ≤ 0.05) between batches at the same day and different capital letters indicate significant differences (*p* ≤ 0.05) between days of ripening at the same batch.

**Table 2 foods-11-02658-t002:** Acids (AU × 10^6^) of “Torta del Casar” cheese during the ripening process.

Origin/Compound	Batches	Days of Ripening				
		0	30	45	60	90
Acids						
Acetic acid	C	5.88 ± 1.114 ^bC^	14.44 ± 5.803 ^aA^	15.44 ± 3.855 ^aA^	16.88 ± 4.768 ^aA^	8.54 ± 0.059 ^aB^
	Lc	10.50 ± 2.972 ^aBC^	16.94 ± 0.681 ^aA^	13.66 ± 1.638 ^aAB^	14.33 ± 3.647 ^aAB^	7.42 ± 0.678 ^aC^
	Lg	7.72 ± 1.183 ^bAB^	14.58 ± 2.117 ^aAB^	16.12 ± 6.524 ^aA^	18.90 ± 6.212 ^aA^	9.25 ± 0.356 ^aC^
Butanoic acid	C	1.86 ± 0.429 ^aB^	3.00 ± 0.258 ^aB^	2.97 ± 0.406 ^bB^	2.99 ± 0.138 ^bB^	5.46 ± 1.380 ^aA^
	Lc	1.88 ± 0.349 ^aB^	3.14 ± 0.143 ^aAB^	2.93 ± 0.450 ^bAB^	3.46 ± 0.649 ^bAB^	4.90 ± 1.446 ^aA^
	Lg	1.95 ± 0.615 ^aC^	3.01 ± 0.012 ^aB^	4.35 ± 0.439 ^aAB^	4.97 ± 0.497 ^aA^	5.45 ± 0.757 ^aA^
Hexanoic acid	C	4.52 ± 0.937 ^aC^	4.44 ± 0.948 ^aC^	8.19 ± 0.618 ^ab2^	9.67 ± 0.778 ^aAB^	11.32 ± 2.618 ^aA^
	Lc	5.06 ± 1.212 ^aB^	7.86 ± 0.986 ^aAB^	8.83 ± 1.899 ^a12^	8.83 ± 0.963 ^aAB^	11.68 ± 4.520 ^aA^
	Lg	5.81 ± 1.496 ^aB^	7.08 ± 0.789 ^aB^	6.84 ± 0.646 ^b2^	9.75 ± 1.75 ^aA^	9.99 ± 1.534 ^aA^
Octanoic acid	C	0.60 ± 0.148 ^bB^	0.38 ± 0.049 ^bB^	1.71 ± 0.313 ^aA^	2.13 ± 0.252 ^aA^	2.34 ± 0.692 ^aA^
	Lc	0.99 ± 0.330 ^aB^	1.62 ± 0.466 ^aAB^	2.60 ± 1.631 ^aAB^	2.57 ± 0.714 ^aAB^	4.64 ± 3.324 ^aA^
	Lg	0.95 ± 0.255 ^aAB^	1.43 ± 0.362 ^aAB^	0.47 ± 0.246 ^aB^	1.62 ± 1.301 ^aAB^	1.94 ±0.493 ^aA^
Propanoic acid	C	n.d.	0.28 ± 0.118 ^bB^	0.97 ± 0.446 ^aAB^	1.53 ± 0.440 ^aA^	0.72 ± 0.529 ^bAB^
	Lc	n.d.	0.88 ± 0.560 ^aA^	1.47 ± 0.955 ^aA^	1.42 ± 0.467 ^aA^	1.45 ± 0.410 ^abA^
	Lg	n.d.	0.25 ± 0.074 ^bB^	0.41 ± 0.155 ^aB^	1.26 ± 0.412 ^aA^	1.71 ± 0.401 ^aA^
2-methyl-propanoic acid	C	0.12 ± 0.002 ^aC^	2.64 ± 1.918 ^aB^	6.01 ± 2.742 ^aA^	6.40 ± 2.368 ^aA^	3.45 ± 0.702 ^bB^
	Lc	0.12 ± 0.002 ^aC^	3.87 ± 1.506 ^aB^	8.21 ± 1.757 ^aAB^	7.18 ± 2.259 ^aAB^	11.94 ± 3.156 ^aA^
	Lg	0.13 ± 0.004 ^aC^	1.02 ± 0.375 ^bC^	6.83 ± 1.688 ^aB^	6.65 ± 2.955 ^aB^	11.02 ± 4.210 ^aA^
3-methyl-butanoic acid	C	2.00 ± 0.265 ^aC^	23.93 ± 1.513 ^aB^	47.41 ± 5.618 ^aA^	54.71 ± 2.382 ^bA^	28.59 ± 4.543 ^bB^
	Lc	0.69 ± 0.113 ^bD^	16.41 ± 2.777 ^aC^	41.59 ± 9.609 ^aBC^	126.91 ± 12.188 ^aA^	79.43 ± 3.102 ^aB^
	Lg	1.33 ± 0.973 ^abC^	10.13 ± 2.629 ^bB^	15.85 ± 2.838 ^bB^	63.26 ± 3.974 ^bA^	61.67 ± 2.608 ^aA^
2-methyl-butanoic acid	C	n.d.	0.93 ± 0.541 ^aB^	3.17 ± 1.271 ^aA^	3.78 ± 0.614 ^aA^	2.75 ± 0.631 ^bA^
	Lc	n.d.	1.37 ± 0.874 ^aC^	2.36 ± 0.771 ^abBC^	5.08 ± 2.271 ^aA^	4.35 ± 0.768 ^aAB^
	Lg	n.d.	0.60 ± 0.145 ^aB^	1.04 ± 0.258 ^bB^	3.11 ± 1.435 ^aAB^	3.83 ± 0.749 ^abA^
3-methyl-2-butenioc acid	C	n.d.	n.d.	0.19 ± 0.040 ^aA^	0.29 ± 0.068 ^aA^	0.17 ± 0.041 ^aA^
	Lc	n.d.	n.d.	0.22 ± 0.058 ^aA^	0.35 ± 0.172 ^aA^	0.20 ± 0.055 ^aA^
	Lg	n.d.	n.d.	0.14 ± 0.002 ^aA^	0.20 ± 0.078 ^aA^	0.16 ± 0.020 ^aA^

Control (uninoculated batch), Lc (batch inoculated with *Lacticaseibacillus casei*) and Lg (batch inoculated with *Lactococcus garvieae*). Values are expressed as mean ± standard deviation. Mean values with different lowercase letters indicate significant differences (*p* ≤ 0.05) between batches at the same day and compound studied. The mean values with different capital letters indicate significant differences (*p* ≤ 0.05) between days at the same batch and compound studied. n.d. not detected.

**Table 3 foods-11-02658-t003:** Alcohols, ketones and aldehydes (AU × 10^6^) of “Torta del Casar” cheese during the ripening process.

Origin/Compound	Batches	Days of Ripening				
		0	30	45	60	90
**Alcohols**						
2-butanol, (R)-	C	n.d.	n.d.	4.27 ± 1.14 ^bB^	12.42 ± 0.860 ^aA^	12.94 ± 1.005 ^aA^
	Lc	n.d.	n.d.	7.58 ± 1.94 ^aA^	5.27 ± 1.198 ^bA^	5.26 ± 0.827 ^bA^
	Lg	n.d.	n.d.	2.21 ± 0.311 ^cB^	10.21 ± 3.759 ^aA^	12.70 ± 1.434 ^aA^
2-methyl-1-propanol	C	0.19 ± 0.014 ^aAB^	0.20 ± 0.011 ^aAB^	0.17 ± 0.031 ^aB^	0.14 ± 0.032 ^aB^	0.25 ± 0.081 ^aA^
	Lc	0.18 ± 0.026 ^aA^	0.17 ± 0.027 ^aA^	0.16 ± 0.038 ^aA^	0.14 ± 0.035 ^aA^	0.14 ± 0.002 ^bA^
	Lg	0.17 ± 0.010 ^aA^	0.18 ± 0.052 ^aA^	0.16 ± 0.044 ^aA^	0.15 ± 0.058 ^aA^	0.17 ± 0.034 ^abA^
3-methyl-1-butanol	C	8.29 ± 0.538 ^aA^	5.70 ± 0.712 ^aB^	3.92 ± 0.426 ^aC^	3.20 ± 0.267 ^aC^	2.97 ± 0.607 ^aC^
	Lc	6.14 ± 0.192 ^bA^	4.63 ± 0.755 ^aB^	4.09 ± 1.013 ^aB^	2.61 ± 0.804 ^aC^	2.57 ± 0.211 ^aC^
	Lg	5.93 ± 0.219 ^bA^	5.14 ± 0.410 ^aA^	3.94 ± 0.818 ^aB^	3.40 ± 0.890 ^aBC^	2.75 ± 0.286 ^aC^
2-methyl-1-butanol	C	n.d.	0.11 ± 0.016 ^aA^	n.d.	n.d.	0.16 ± 0.042 ^aA^
	Lc	n.d.	n.d.	n.d.	0.12 ± 0.022 ^aA^	0.21 ± 0.015 ^aA^
	Lg	n.d.	n.d.	0.12 ± 0.009 ^aA^	0.11 ± 0.000 ^aA^	0.15 ± 0.027 ^aA^
Phenylethyl alcohol	C	1.36 ± 0.137 ^aA^	0.86 ± 0.094 ^bC^	1.04 ± 0.151 ^aBC^	1.02 ± 0.173 ^aAB^	0.93 ± 0.089 ^aC^
	Lc	1.25 ± 0.039 ^aA^	1.08 ± 0.075 ^aA^	1.07 ± 0.308 ^aA^	0.99 ± 0.145 ^aA^	0.89 ± 0.207 ^aA^
	Lg	1.30 ± 0.404 ^aA^	1.21 ± 0.180 ^aAB^	0.92 ± 0.164 ^aAB^	0.97 ± 0.149 ^aAB^	0.80 ± 0.149 ^aB^
2,3-butanediol, [R-(R*,R*)]-	C	2.80 ± 0.853 ^aB^	15.42 ± 1.458 ^aA^	10.01 ± 1.038 ^bA^	9.59 ± 1.204 ^abAB^	10.93 ± 0.449 ^aA^
	Lc	2.54 ± 0.822 ^aB^	14.50 ± 1.181 ^aA^	4.20 ± 0.973 ^cAB^	6.19 ± 0.959 ^bAB^	3.09 ± 0.144 ^bB^
	Lg	2.58 ± 0.738 ^aC^	14.32 ± 1.674 ^aAB^	14.81 ± 0.634 ^aA^	12.59 ± 1.500 ^aA^	5.77 ± 0.528 ^bBC^
2-butoxy-ethanol	C	n.d.	0.27 ± 0.054 ^aA^	0.20 ± 0.089 ^bAB^	0.15 ± 0.021 ^bB^	0.23 ± 0.083 ^bAB^
	Lc	n.d.	0.30 ± 0.056 ^aA^	0.24 ± 0.082 ^bAB^	0.20 ± 0.033 ^bB^	0.22 ± 0.039 ^bAB^
	Lg	n.d.	0.32 ± 0.052 ^aB^	0.39 ± 0.036 ^aA^	0.37 ± 0.041 ^aA^	0.36 ± 0.042 ^aA^
2,6-dimethyl-4-heptanol	C	n.d.	0.26 ± 0.004 ^aA^	0.37 ± 0.462 ^aA^	0.19 ± 0.022 ^aA^	0.18 ± 0.037 ^aA^
	Lc	n.d.	0.19 ± 0.017 ^aB^	0.66 ± 0.146 ^aA^	0.28 ± 0.159 ^aB^	0.37 ± 0.029 ^aB^
	Lg	n.d.	0.24 ± 0.023 ^aA^	0.35 ± 0.211 ^aA^	0.18 ± 0.048 ^aA^	0.19 ± 0.015 ^aA^
**Ketones**						
2-nonanone	C	0.38 ± 0.086 ^abB^	0.14 ± 0.038 ^aB^	0.41 ± 0.339 ^aB^	0.19 ± 0.002 ^aB^	0.61 ± 0.021 ^bA^
	Lc	0.54 ± 0.182 ^aB^	0.13 ± 0.006 ^aB^	0.21 ± 0.089 ^aB^	0.18 ± 0.056 ^aB^	0.95 ± 0.050 ^aA^
	Lg	0.23 ± 0.028 ^bB^	0.16 ± 0.036 ^aB^	0.18 ± 0.051 ^aB^	0.11 ± 0.006 ^aB^	0.79 ± 0.152 ^abA^
2-heptanone	C	0.32 ± 0.009 ^aAB^	0.12 ± 0.032 ^aB^	0.26 ± 0.078 ^aAB^	0.16 ± 0.052 ^aB^	5.23 ± 2.880 ^aA^
	Lc	0.38 ± 0.138 ^aAB^	0.12 ± 0.004 ^aB^	0.19 ± 0.091 ^aAB^	0.18 ± 0.060 ^aAB^	1.10 ± 0.096 ^bA^
	Lg	0.27 ± 0.057 ^aB^	0.13 ± 0.004 ^aB^	0.28 ± 0.155 ^aB^	0.13 ± 0.009 ^aB^	0.98 ± 0.147 ^bA^
2,3-butanedione	C	0.25 ± 0.040 ^aC^	0.57 ± 0.104 ^aABC^	0.88 ± 0.322 ^aA^	0.72 ± 0.221 ^aAB^	0.40 ± 0.115 ^abBC^
	Lc	0.27 ± 0.066 ^aC^	0.43 ± 0.075 ^abC^	1.00 ± 0.128 ^aA^	0.69 ± 0.140 ^aB^	0.34 ± 0.080 ^bC^
	Lg	0.26 ± 0.079 ^aC^	0.39 ± 0.104 ^bBC^	1.16 ± 0.202 ^aA^	0.56 ± 0.066 ^aB^	0.41 ± 0.037 ^aBC^
2-pentanone	C	0.16 ± 0.038 ^aA^	0.12 ± 0.012 ^aA^	0.15 ± 0.004 ^aA^	n.d.	1.27 ± 0.103 ^aA^
	Lc	n.d.	n.d.	n.d.	n.d.	0.87 ± 0.098 ^aA^
	Lg	n.d.	n.d.	0.16 ± 0.095 ^aA^	n.d.	0.40 ± 0.021 ^aA^
2-butanone	C	0.30 ± 0.099 ^aD^	1.12 ± 0.626 ^aD^	10.17 ± 1.942 ^aC^	18.70 ± 1.575 ^aB^	35.37 ± 4.673 ^aA^
	Lc	0.31 ± 0.096 ^aD^	1.13 ± 0.403 ^aD^	9.62 ± 1.403 ^aC^	15.76 ± 1.897 ^bB^	25.99 ± 3.316 ^bA^
	Lg	0.28 ± 0.117 ^aD^	0.53 ± 0.220 ^aD^	9.30 ± 3.246 ^aC^	17.64 ± 1.161 ^abA^	31.63 ± 1.967 ^abA^
**Aldehydes**						
3-methyl-butanal	C	0.17 ± 0.020 ^aB^	0.38 ± 0.112 ^aAB^	0.34 ± 0.142 ^abAB^	0.61 ± 0.202 ^aA^	0.60 ± 0.049 ^aAB^
	Lc	0.17 ± 0.020 ^aB^	0.33 ± 0.151 ^aAB^	0.55 ± 0.100 ^aA^	0.41 ± 0.185 ^bAB^	0.69 ± 0.027 ^aA^
	Lg	0.16 ±0.013 ^aB^	0.22 ± 0.039 ^aB^	0.21 ± 0.088 ^bB^	0.31 ± 0.155 ^bB^	0.68 ± 0.130 ^aA^

Control (uninoculated batch), Lc (batch inoculated with *Lacticaseibacillus casei*) and Lg (batch inoculated with *Lactococcus garvieae*). Values are expressed as mean ± standard deviation. Mean values with different lowercase letters indicate significant differences (*p* ≤ 0.05) between batches at the same day and compound studied. The means with different capital letters indicate significant differences (*p* ≤ 0.05) between days at the same batch and compound studied. n.d. not detected.

**Table 4 foods-11-02658-t004:** Esters and others volatile compounds (AU × 10^6^) of “Torta del Casar” cheese during the ripening.

Origin/Compound	Batches	Days of Ripening				
		0	30	45	60	90
**Esters**						
Butanoic acid, ethyl ester	C	1.67 ± 0.156 ^aA^	0.40 ± 0.151 ^aB^	0.17 ± 0.027 ^aBC^	0.12 ± 0.003 ^aC^	0.14 ± 0.002 ^aC^
	Lc	1.69 ± 0.058 ^aA^	0.24 ± 0.028 ^bB^	0.15 ± 0.024 ^aC^	0.18 ± 0.061 ^aBC^	0.13 ± 0.017 ^aC^
	Lg	2.00 ± 0.449 ^aA^	0.26 ± 0.089 ^abB^	0.21 ± 0.052 ^aB^	0.19 ± 0.046 ^aB^	0.15 ± 0.010 ^aB^
Hexanoic acid, ethyl ester	C	2.57 ± 0.439 ^aA^	0.79 ± 0.233 ^aB^	0.54 ± 0.069 ^abB^	0.43 ± 0.042 ^aB^	0.53 ± 0.090 ^aB^
	Lc	2.08 ± 0.057 ^aA^	0.57 ± 0.058 ^aB^	0.40 ± 0.067 ^bC^	0.46 ± 0.148 ^aBC^	0.32 ± 0.032 ^bC^
	Lg	2.61 ± 0.856 ^aA^	0.69 ± 0.130 ^aB^	0.66 ± 0.158 ^aB^	0.58 ± 0.113 ^aB^	0.56 ± 0.050 ^aB^
Octanoic acid, ethyl ester	C	1.85 ± 0.305 ^aA^	0.40 ± 0.087 ^aB^	0.41 ± 0.030 ^aB^	0.41 ± 0.018 ^aB^	0.49 ± 0.053 ^aB^
	Lc	1.60 ± 0.134 ^aA^	0.42 ± 0.044 ^aB^	0.40 ± 0.041 ^aB^	0.35 ± 0.040 ^bB^	0.35 ± 0.028 ^bB^
	Lg	1.97 ± 0.757 ^aA^	0.42 ± 0.038 ^aB^	0.36 ± 0.055 ^aB^	0.36 ± 0.032 ^abB^	0.45 ± 0.040 ^aB^
Decanoic acid, ethyl ester	C	2.18 ± 0.258 ^bA^	0.56 ± 0.135 ^aB^	0.62 ± 0.053 ^aB^	0.65 ± 0.030 ^aB^	0.64 ± 0.073 ^aB^
	Lc	2.79 ± 0.577 ^abA^	0.65 ± 0.075 ^aB^	0.68 ± 0.102 ^aB^	0.53 ± 0.044 ^bB^	0.45 ± 0.094 ^bB^
	Lg	2.97 ± 0.856 ^aA^	0.62 ± 0.065 ^aB^	0.48 ± 0.058 ^bB^	0.52 ± 0.051 ^bB^	0.56 ± 0.111 ^abB^
Dodecanoic acid, ethyl ester	C	0.18 ± 0.025 ^a^	n.d.	n.d.	n.d.	n.d.
	Lc	0.21 ± 0.044 ^a^	n.d.	n.d.	n.d.	n.d.
	Lg	0.24 ± 0.069 ^a^	n.d.	n.d.	n.d.	n.d.
1-butanol, 3-methyl-, acetate	C	0.37 ± 0.015 ^aA^	0.37 ± 0.035 ^aA^	0.28 ± 0.043 ^abAB^	0.23 ± 0.042 ^aB^	0.33 ± 0.093 ^aAB^
	Lc	0.35 ± 0.039 ^aA^	0.27 ± 0.065 ^bAB^	0.18 ± 0.025 ^bB^	0.19 ± 0.095 ^aB^	0.19 ± 0.025 ^bB^
	Lg	0.37 ± 0.027 ^aA^	0.27 ± 0.046 ^bA^	0.34 ± 0.087 ^aA^	0.30 ± 0.099 ^aA^	0.35 ± 0.025 ^aA^
Others compounds						
1,5,9-decatriene, 2,3,5,8-tetramethyl-	C	0.60 ± 0.089 ^bC^	0.61 ± 0.108 ^bC^	0.77 ± 0.048 ^abAB^	0.89 ± 0.025 ^aA^	0.67 ± 0.023 ^aBC^
	Lc	0.71 ± 0.084 ^abAB^	0.85 ± 0.068 ^aA^	0.85 ± 0.073 ^aA^	0.70 ± 0.051 ^bAB^	0.64 ± 0.133 ^aB^
	Lg	0.76 ± 0.017 ^aAB^	0.80 ± 0.053 ^aA^	0.68 ± 0.032 ^bB^	0.74 ± 0.048 ^bAB^	0.66 ± 0.089 ^aB^
Dimethyl ether	C	29.18 ± 3.598 ^aA^	5.55 ± 3.239 ^aB^	1.39 ± 0.562 ^aBC^	1.18 ± 0.068 ^abBC^	0.66 ± 0.278 ^bBC^
	Lc	31.91 ± 1.945 ^aA^	3.78 ± 0.655 ^aB^	1.30 ± 0.699 ^aC^	0.41 ± 0.075 ^bC^	0.25 ± 0.093 ^cC^
	Lg	31.42 ± 0.338 ^aA^	3.50 ± 1.693 ^aB^	1.63 ± 0.826 ^aC^	1.75 ± 0.689 ^aBC^	1.50 ± 0.080 ^aC^
Trimethyl-pyrazine	C	n.d.	n.d.	0.13 ± 0.009 ^B^	0.14 ± 0.012 ^B^	0.42 ± 0.129 ^abA^
	Lc	n.d.	n.d.	n.d.	n.d.	0.68 ± 0.099 ^a^
	Lg	n.d.	n.d.	n.d.	n.d.	0.28 ± 0.081 ^b^
2,5-dimethyl-pyrazine	C	n.d.	n.d.	n.d.	n.d.	0.18 ± 0.007 ^a^
	Lc	n.d.	n.d.	n.d.	n.d.	0.37 ± 0.054 ^a^
	Lg	n.d.	n.d.	n.d.	n.d.	0.15 ± 0.008 ^a^
Dimethyl disulfide	C	0.15 ± 0.020 ^aB^	0.52 ± 0.085 ^aA^	0.36 ± 0.094 ^aAB^	0.21 ± 0.038 ^aB^	0.17 ± 0.006 ^aB^
	Lc	n.d.	0.50 ± 0.089 ^aA^	0.35 ± 0.011 ^aAB^	0.29 ± 0.022 ^aB^	0.16 ± 0.023 ^aB^
	Lg	n.d.	0.27 ± 0.044 ^bA^	0.16 ± 0.039 ^bB^	0.28 ± 0.082 ^aA^	0.14 ± 0.002 ^aB^

Control (uninoculated batch), Lc (batch inoculated with *Lacticaseibacillus casei*) and Lg (batch inoculated with *Lactococcus garvieae*). Values are expressed as mean ± standard deviation. Mean values with different lowercase letters indicate significant differences (*p* ≤ 0.05) between batches at the same day and compound studied. The means with different capital letters indicate significant differences (*p* ≤ 0.05) between days at the same batch and compound studied. n.d. not detected.

**Table 5 foods-11-02658-t005:** Values of instrumental texture (hardness, adhesiveness, springiness, cohesiveness and chewiness) and color parameters of “Torta del Casar” cheese at the end of the ripening process.

Parameters	Batches		
	C	Lc	Lg
Hardness (N)	5.05 ± 2.663	5.18 ± 1.579	4.44 ± 1.702
Adhesiveness (N/s)	−0.36 ± 0.192	−0.55 ± 0.312	−0.47 ± 0.262
Springiness	0.74 ± 0.052	0.76 ± 0.068	0.75 ± 0.079
Cohesiveness	0.64 ± 0.049	0.64 ± 0.033	0.65 ± 0.039
Chewiness (N)	2.66 ± 0.831	2.57 ± 0.885	2.54 ± 0.828
CIE L*	101.09 ± 6.250	100.51 ± 2.730	97.11 ± 3.740 *
CIE a*	−1.43 ± 0.859	−1.26 ± 0.920	−1.72 ± 0.640
CIE b*	5.49 ± 2.510	6.26 ± 1.190	5.17 ± 2.211

C (uninoculated batch), Lc (inoculated with *Lacticaseibacillus casei*) and Lg (inoculated with *Lactococcus garvieae*). Values are expressed as mean ± standard deviation. Asterisks indicate significant differences with respect to control batch.

## Data Availability

Data is contained within the article.

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
