# Peer review of "Evolution of Volatile Compounds during Ripening and Final Sensory Changes of Traditional Raw Ewe’s Milk Cheese “Torta del Casar” Maturated with Selected Protective Lactic Acid Bacteria"

_foods, 2022, doi:10.3390/foods11172658_

Round 1

Reviewer 1 Report

The current manuscript in general deals with an interesting and relevant topic and contain some noticeable results, but, unfortunately, the manuscript is not suitable for publication in the present form. The findings from this study are interesting and will help researchers working in the field, however, there are some issues listed in attached pdf file need to be addressed before publication.

Author Response

Reviewer 1

The current manuscript in general deals with an interesting and relevant topic and contain some noticeable results, but, unfortunately, the manuscript is not suitable for publication in the present form. The findings from this study are interesting and will help researchers working in the field, however, there are some issues listed in attached pdf file need to be addressed before publication.

Response:

We thank Reviewer 1 for the positive comments on the manuscript.

There are some issues listed in attached pdf file need to be addressed before publication.

The source of all reagents, bacteria and equipment used should be stated. Add in all materials and methods

Response:

The source of all reagents, bacteria and equipment has been included in the revised version of the manuscript.

Line 51- Please write here an appropriate abbreviation according to the new nomenclature. This is not Lactococcus casei but formerly Lactobacillus casei and currently Lacticaseibacillus casei.

Response:

As indicated by Reviewer 1, the proper abbreviation has been used and  revised throughout the manuscript.

Line 51- The correct abbreviation is L. or Lc. Correct in all manuscript.

Response:

This been done and revised in the new version of the manuscript.

Line 56- add space

Response:

A space has been included.

Line 61- add country and city

Response:

Country and city have been added in the revised version.

Line 64- source

Response:

The source has been added in the new version of the manuscript.

Line 84- Add correctly spelled Celsius symbol

Response:

Celsius symbol has been corrected.

Line 93- Add under what conditions anaerobic? are they relatively anaerobic?

Response:

The PCA medium was incubated under aerobic conditions and the MRS medium was incubated under microaerophilic conditions. This has been included in the revised version of the manuscript.

Line 94- Add under what conditions?

Response:

The Enterobacteriaceae were incubated under aerobic conditions. This has been added in the new version of the manuscript.

Line 104- Please provide the exact TPA measurement parameters, type of stylus used, measuring head movement speed, whether the parameters were determined on the basis of % deformation or other.

Response:

According to Reviewer 1 requirements, all the TPA measurement parameters have been included in the Material and Methods section of the revised manuscript.

Line 120- Add source of all reagents.

Response:

The source of all reagents has been included in the Material and Methods section of the revised manuscript. 

Line 165- This is not marked on the chart. If we write a statistical analysis on a graph, the same letters or numbers should also be found for the mean values in which we do not find differences.

Response:

As requested by Reviewer 1, the same letters have been introduced for the mean values in which we did not find differences.

Line 171- space

Response:

A space has been entered

Line 182- As we give the differences between the means on the chart, the markings should also be included if we do not find these differences.

Only then will it be consistent with the text of the work being read.

Response:

In the new version of the manuscript, equal marks have been included in those means in which we did not find differences.

Line 185- Figure 1 is difficult to understand in its present form as far as the recording of the statistical analysis is concerned. I propose to write the figure 1 graph under the graph so that the individual graphs are a bit larger.

Or save the results in the form of a table so that there is no doubt about it.

Response:

The purpose of Figure 1 is to show to the readers the evolution of moisture content, water activity and pH throughout the cheese ripening process and possible significant differences between batches on the same day of analysis. All these aspects are clearly shown in the revised version of Figure 1. The significant differences between days have been indicated in the text but not in the Figure for a better vision and understanding of that.    

Line 203- Complete the statistical analysis - in the table, add the same letters and numbers where there were no differences. The same applies to the other Tables.

Response:

As the Reviewer 1 suggests, same letters (lower and upper case) have been added for the mean values in which we did not found differences. This has been modified in all tables and figures of the revised version.

Line 218- This is not in Table 2.

Response:

This has been revised and corrected in Table 2 in the new version of the manuscript.

Line 240-241- Compare the notation of acids in the text and how they are written in the Table. No consistency.

Response:

This has been revised and corrected in Table 2 in the new version of the manuscript.

Line 251- Save correctly

Response:

Table 2 has been corrected in the new version of the manuscript

The discussion does not fit together. In the marked lines 282-285 and 326-328 it is written that these bacterial cultures have no influence on these compounds in the tested cheeses and here their positive effect is determined, and I wonder how? Another example is line 338-339 where it was written that only one compound showed a difference?

Perhaps I did not understand it, but it seems to me that in some parts there is a lack of precision and correctness of the formulated conclusions. Unambiguous statements should be avoided.

Response:

According to the Reviewer 1, the unambiguous statements have deleted or rewritten in the previous version to enhance the quality of the discussion of the revised manuscript.

Line 394- We cannot say at the end of the article that these strains of bacteria can be used as protective in cheese. The conclusions that arise are that they did not affect the formation of undesirable taste and flavor defects in cheeses and the changes in physical, chemical and textural characteristics, as well as the number of Enterobacteriacee bacteria. And how they will be protective in cheeses was not the subject of this study. In addition, they did not reduce the number of Enterobacteriaceae at the end of cheese ripening compared to the control cheeses without their addition. So, after these studies, we are not able to state what effect these bacterial strains will have on the microorganisms in the cheese.

The conclusion is that they did not change any of the characteristics of the cheeses except that could have a positive effect on the aroma, for their contribution to the generation of methyl branched compounds (mainly methyl branched acids) and for not increasing and even reducing oxidation compounds from β-oxidation of fatty acids. Please correct the conclusions.

Response:

Reviewer 1 is right, for this reason the conclusions of the manuscript have been rewritten and the sentence “Thus, these microorganisms could be proposed as protective culture in soft-ripened “Torta del Casar” cheese” has been removed.

Reviewer 2 Report

The manuscript entitled " Evolution of volatile compounds during ripening and final sensory changes of traditional raw ewe's milk cheese “Torta del  Casar” maturated with selected protective lactic acid bacteria”  aims to evaluate the effect of selected LAB  strains on the evolution of the volatile compounds throughout the ripening process and the final sensory traits of “Torta del Casar” cheese.

There are many points on which I think authors should provide more explanatory answers.

The design of the experimental process and the parameters examined are not satisfactory to answer key questions.

More specific:

I believe that more literature review is needed regarding the biochemical and technological properties of the 2 protective bacterial cultures (Lc. casei 116 and Lco. garvieae 151)

Τhe mode of incorporation of the inoculum raises many questions, both in terms of proper use and effectiveness

(the inoculation was done in the centre of the curd (in a cube of 16 cm2 of surface and 6 cm deep)

We consider that it is necessary to evaluate the viability of 2 protective cultures during cheese ripening, to claim that protective cultures contribute to the differentiation of volatile aromatic components in cheeses.

This rather essential as the cheese is made from raw milk. Also, the use of vegetable rennet causes extensive proteolysis resulting in the formation of various volatile components, making the interpretation of volatile analysis results even more difficult.

We also consider that the determination of total aerobic microorganism (TAM), lactic acid bacteria (LAB), and Enterobacteriaceae (E) are not capable of proving that the eventual changes both in the aromatic components and the volatile compounds throughout the ripening process and the final sensory traits of “Torta del Casar” cheese are due to the use of protective cultures. (why?) We also consider that the determination of other categories of cheese microflora (yeats, molds) and mainly the control of the presence of pathogenic microorganisms Staph. aureus, L. monocytogenes is necessary to be determined.

We consider that solely the determination of parameters such as ph, humidity, and aw, are not sufficient to interpret the changes that lead to the formation of the aroma profile of the cheese. Some other parameters in addition, such as salt content, fat content, as well as nitrogen fraction, could provide additional information for the interpretation of the results.

My general impression is that the manuscript in its current state is weak, with substantial limitations.

Author Response

Reviewer 2

The manuscript entitled " Evolution of volatile compounds during ripening and final sensory changes of traditional raw ewe's milk cheese “Torta del Casar” maturated with selected protective lactic acid bacteria” aims to evaluate the effect of selected LAB strains on the evolution of the volatile compounds throughout the ripening process and the final sensory traits of “Torta del Casar” cheese.

There are many points on which I think authors should provide more explanatory answers.

The design of the experimental process and the parameters examined are not satisfactory to answer key questions.

More specific:

I believe that more literature review is needed regarding the biochemical and technological properties of the 2 protective bacterial cultures (Lc. casei 116 and Lco. garvieae 151)

Response:

More information about the 2 protective bacterial L. casei and Lc. garvieae has been included in the Introduction section of the revised manuscript.

Τhe mode of incorporation of the inoculum raises many questions, both in terms of proper use and effectiveness (the inoculation was done in the centre of the curd (in a cube of 16 cm2 of surface and 6 cm deep).

Response:

To avoid contamination in the industry with the two assayed LAB strains (L. casei and Lc. garvieae), since their industrial use is not at moment allowed, the “Torta del Casar” cheeses used in this study were first elaborated in a cheese factory located in Cáceres (Extremadura region, Spain). Next, on the first day of processing, cheeses were transported to the pilot plant of the Faculty of Veterinary under refrigerated conditions,  inoculated and ripened for up to 90 days. The inoculation was done in the center of the curd (in a cube of 16 cm2 of surface and 6 cm deep). For the microbiological analysis, the entire cube was used in all sampling times. This has been included in the revised version of the manuscript.

We consider that it is necessary to evaluate the viability of 2 protective cultures during cheese ripening, to claim that protective cultures contribute to the differentiation of volatile aromatic components in cheeses.

Response:

The evaluation of the implantation of L. casei 116 and Lc. garvieae 151 in the batches Lc and Lg was done in MRS plates in the last sampling time (90 days) following the procedure described by Martín et al. (2021). The identification of the LAB strains was performed by sequencing analysis of the 16S rRNA region according to the methodology proposed by Walter et al. (2000) and PFGE analysis of the DNA following procedures previously described by Alía et al. (2020). Most of the investigated o LAB isolates of bathes Lc and Lg were identified as L. casei 116 and Lc. garvieae 151, respectively, by sequencing analysis of the 16S rRNA region and PFGE analysis, confirming the implantation of these strains in the inoculated batches. This paragraph has been included in the revised version of the manuscript.

This rather essential as the cheese is made from raw milk. Also, the use of vegetable rennet causes extensive proteolysis resulting in the formation of various volatile components, making the interpretation of volatile analysis results even more difficult.

Response:

We agree with Reviewer 2 that it is difficult the interpretation of volatile analysis results in cheeses made with raw milk and vegetable rennet causing of extensive proteolysis. For this reason, the interpretation of volatile analysis is conducted taken into account the differences between the control and inoculated batches since the only difference between the control and inoculated batch is the presence in the last ones of LAB strains.

We also consider that the determination of total aerobic microorganism (TAM), lactic acid bacteria (LAB), and Enterobacteriaceae (E) are not capable of proving that the eventual changes both in the aromatic components and the volatile compounds throughout the ripening process and the final sensory traits of “Torta del Casar” cheese are due to the use of protective cultures. (why?) We also consider that the determination of other categories of cheese microflora (yeats, molds) and mainly the control of the presence of pathogenic microorganisms Staph. aureus, L. monocytogenes is necessary to be determined.

Response:

The main microbial groups in “Torta del Casar” cheese (TAM, LAB and Enterobacteriaceae were determined). The main difference between the control and inoculated batches is the presence of the inoculated L. casei 116 and Lc. garvieae 151 in the LAB population. Yeast is detected in “Torta del Casar” at low levels and moulds grow on the surface of the product and practically are not detected in this kind of product. Thus, the difference in volatile compounds between control and inoculated batches is due to the presence of these inoculated LAB strains. In addition, the effectivity of the inoculated strains in the inhibition of the L. monocytogenes during ripening has been previously demonstrated and this is not the purpose of this work. For this reason, this pathogen was not determined.  

We consider that solely the determination of parameters such as ph, humidity, and aw, are not sufficient to interpret the changes that lead to the formation of the aroma profile of the cheese. Some other parameters in addition, such as salt content, fat content, as well as nitrogen fraction, could provide additional information for the interpretation of the results.

Response:

We agree with Reviewer 2 that the pH, humidity, and water activity are not sufficient to interpret the changes that lead to the formation of the aroma profile of the cheese. But the humidity and aw are the parameters that could influence the growth of the inoculated LAB strains and pH could be influenced by the growth of LAB strains. For this reason, these parameters were determined while others like fat, nitrogen fraction and salt content were not since no differences between control and inoculated batches are expected.    

My general impression is that the manuscript in its current state is weak, with substantial limitations.

Response:

We think that the revised manuscript has been greatly improved with the modifications and suggestions of the reviewers that have been included in the revised version of the manuscript. In our opinion, the revised version of the manuscript offers interesting information to the reader of Foods and could be now suitable for publishing in this Journal.

Reviewer 3 Report

The topic of the article falls within the thematic scope of the journal FOODS.

The purpose of this study was to evaluate the effect of the addition of selected two LAB strains (previously shown to be active against L. monocytogenes) isolated from raw ewe´s milk cheeses on the evolution of the volatile compounds throughout the ripening process and the final sensory traits of traditional “Torta del Casar” cheese.

I have few comments to the manuscript - all suggestions for corrections were introduced in the review mode to the attached pdf file:

- Chapter 2.5 and 2.8 require supplementing - lack of necessary methodological details

- In Table 2, errors in the nomenclature of the tested compounds

- Throughout the manuscript, a “terrible”, illegible way of presenting the results of statistical analysis - I propose to replace it with a well-known and commonly used system with lower- and uppercase letters of homogeneous groups.

- The final conclusion (lines 394-395) is beyond the scope of the research carried out.

 After taking these changes into account, the manuscript may be subject to further editorial work.

Author Response

Reviewer 3

The topic of the article falls within the thematic scope of the journal FOODS.

The purpose of this study was to evaluate the effect of the addition of selected two LAB strains (previously shown to be active against L. monocytogenes) isolated from raw ewe´s milk cheeses on the evolution of the volatile compounds throughout the ripening process and the final sensory traits of traditional “Torta del Casar” cheese.

Response:

We thank Reviewer 3 for these positive comments.

I have few comments to the manuscript - all suggestions for corrections were introduced in the review mode to the attached pdf file:

- Chapter 2.5 and 2.8 require supplementing - lack of necessary methodological details

- In Table 2, errors in the nomenclature of the tested compounds

Response:

Reviewer 3 is right. This has been amended in the corrected version of the manuscript.

- Throughout the manuscript, a “terrible”, illegible way of presenting the results of statistical analysis - I propose to replace it with a well-known and commonly used system with lower- and uppercase letters of homogeneous groups.

Response:

As reviewer 3 proposes, lowercase and capital letters have been used to show the results of the statistical analyses.

- The final conclusion (lines 394-395) is beyond the scope of the research carried out.

Response:

The conclusion has been rewritten and the final sentence (lines 394-395) has been removed.

After taking these changes into account, the manuscript may be subject to further editorial work.

Response:

We think that the quality of the revised manuscript has been greatly improved with the modifications and suggestions of the reviewers that have been included in the revised version. In our opinion, the revised version of the manuscript offers interesting information to the reader of Foods and could be now suitable for publishing in this Journal.

Line 91-whether incubation on the MRS was aerobic or anaerobic?

Response:

The incubation on the MRS was under microaerophilic conditions. This has been included in the new version of the manuscript.

Line 91-Why was the PCA medium used and not the PCSMA medium suggested for dairy products - i.e. with the addition of milk powder?

Response

We agree with Reviewer 3 that probably PCSMA could be better than PCA, but we used this medium because it has been previously successfully reported in “Torta del Casar” cheese and other types of cheeses for total aerobic bacteria counts analysis (Ordiales et al. 2013. Food Control 33, 448-454; Rodríguez-Pinilla et al., 2015. Dairy Sci. & Technol. (2015) 95:425–436)

Line 111-complete the remaining information on the method used

Response

Information about the TPA measurement parameters has been included in the Material and Methods section of the revised manuscript.

Line 156-complete the remaining information on the method used; there is no information here what method was used, what discriminants were assessed, or a point scale?

Response

The remaining information on the triangular olfactory test for the sensory evaluation has been included in the Material and Methods section of the revised manuscript.

Line 160, 177- italic

Response:

This has been corrected in the new version of the manuscript.

Line 181- figure 1-first remark: such marks of the results of statistical analysis are completely illegible, I suggest introducing the commonly used method - that is, upper and lower case letters; second note: why are not all the results of the statistical analysis only selected?

Response:

For a better understanding, only differences between batches on the same day have been included in Figure 1. The significant differences of data between days in this Figure have been commented in the text.

Line 251- Table 2- ??? are the same acids? it appears from the text that it is not

Response:

As suggested by Reviewers 1 and 3, this has been amended in the new version of the manuscript.

Line 367-rhamnosus

Response:

This has been amended in the new version.

Line 394-395- it was not tested at work - the conclusion is beyond the scope of the research carried out.

Response:

Reviewer 3 is right. The conclusion has been rewritten and this sentence has been deleted in the new version of the manuscript.

Reviewer 4 Report

The paper under consideration intends to evaluate the impact of two protective LAB cultures in Torta del Casar cheese, a very interesting topic, especially in cases where the authenticity of the origin must be ensured.

It is based on adequate analysis methodologies, although some other chemical parameters like fat, protein and proteolysis indexes could be included, and with interesting results regarding the objectives of the study. However, the conclusions must not be completely accepted because the study apparently suffers from a major limitation according to my reading. I could not find the number of repetitions of the experimental protocol, which leads me to suppose that the study was based only on a technological test, without test replicas, which is not acceptable. If this is true (in the text I do not find any reference to the number of test replicates) it is not acceptable, the standard deviations that are shown refer only to the analytical determinations and therefore the results should not be interpreted in a generalized way. For this purpose, a sufficient number of repetitions of the tests must be carried out, in my opinion at least three replicas, so that the interpretation of the results can be correctly carried out.

Despite this, the results are well presented and discussed, although Introduction fail to analyse the although the introduction fails to discuss the regulatory implications of using added cultures in the manufacture of PDO Torta del Casar cheese. Consequently, I think the introduction should be revised.

These are the most relevant generic comments about the submitted paper. Some detailed suggestions are as follows

Suggestions:

1. Introduction - The introduction must be revised according to the above mentioned.

2.2. Preparation of cheese – How inoculation is made, in granny curd or in pressed curd block? Please better explain the inoculation process and discuss and discuss its effectiveness.

2.2. Preparation of cheese – It is unclear how many times (how many replicates) the experimental design was repeated. It seems that there is only one trial. If so, it is not adequate for the discussion and the generalization of the results. The trial protocol should be repeated at least three times.

2.4. Physicochemical analysis – If there are available, some other chemical parameters like fat, protein and proteolysis indexes should be included as they are more important for cheese quality than colour, for instance.

Lines 226 – the authors mentioned that the decrease of Enterobacteriaceae throughout the ripening is probably due to antimicrobial effect of the dominant LAB population. This was observed in all batches including the control. This mean that pH, water content and aw decreases has no effect at all? Please discuss this.

Author Response

The paper under consideration intends to evaluate the impact of two protective LAB cultures in Torta del Casar cheese, a very interesting topic, especially in cases where the authenticity of the origin must be ensured.

Response:

We thank Reviewer 4 for the positive comments on the manuscript.

It is based on adequate analysis methodologies, although some other chemical parameters like fat, protein and proteolysis indexes could be included, and with interesting results regarding the objectives of the study. However, the conclusions must not be completely accepted because the study apparently suffers from a major limitation according to my reading. I could not find the number of repetitions of the experimental protocol, which leads me to suppose that the study was based only on a technological test, without test replicas, which is not acceptable. If this is true (in the text I do not find any reference to the number of test replicates) it is not acceptable, the standard deviations that are shown refer only to the analytical determinations and therefore the results should not be interpreted in a generalized way. For this purpose, a sufficient number of repetitions of the tests must be carried out, in my opinion at least three replicas, so that the interpretation of the results can be correctly carried out.

Response:

The experiment consisting of 3 different batches x 5 sampling times x 5 different cheese/each batch and sampling time, which were evaluated once, according to the European Union Reference Laboratory Technical Guidance Document to determine the influence of selected LAB strains in shelf-life studies on L. monocytogenes in RTE foods (such as “Torta del Casar” cheese) where no growth or the growth probability of this pathogen is ≤ 10 % (Beaufort et al., 2014). This paragraph has been included in the revised version of the manuscript.

Furthermore, processing of “Torta del Casar” cheese was followed according to the industrial procedure for this product, using raw milk from at least five different farms and evolution of the physicochemical parameters was consistent with those observed in previous published papers of this kind of cheese (Ordiales et al., 2013; 2014; Crespo et al., 2022). Thus, it is not expected in “Torta del Casar” cheese processed under industrial conditions, using raw milk from different farms, that successive repetitions of the entire experiment (with the abovementioned batches, sampling times and cheeses), could lead to results different from those discussed in the present work.

Despite this, the results are well presented and discussed, although Introduction fail to analyse the although the introduction fails to discuss the regulatory implications of using added cultures in the manufacture of PDO Torta del Casar cheese. Consequently, I think the introduction should be revised.

These are the most relevant generic comments about the submitted paper. Some detailed suggestions are as follows

Suggestions:

  1. Introduction - The introduction must be revised according to the above mentioned.

Response:

Despite the advantages that the use of protective cultures of LAB could suppose for safety control of “Torta del Casar” cheese, to date, the use of protective cultures in this kind of cheese has not been reported in the literature. This is the first paper where selected protective cultures of LAB strains are evaluated to be used in this kind of traditional  cheese. This has been included in the Introduction section of the revised version of the manuscript.

2.2. Preparation of cheese – How inoculation is made, in granny curd or in pressed curd block? Please better explain the inoculation process and discuss and discuss its effectiveness.

Response:

On the first day of processing, cheeses were transported to the pilot plant of the Faculty of Veterinary under refrigerated conditions, inoculated and ripened for up to 90 days. The inoculation was done in the center of the curd (in a cube of 16 cm2 of surface and 6 cm deep). For all the analysis, the entire cube was used in all sampling times. This has been included in the revised version of the manuscript.

2.2. Preparation of cheese – It is unclear how many times (how many replicates) the experimental design was repeated. It seems that there is only one trial. If so, it is not adequate for the discussion and the generalization of the results. The trial protocol should be repeated at least three times.

Response:

The experiment consisting of 3 different batches x 5 sampling times x 5 different cheese/each batch and sampling time, which were evaluated once, according to the European Union Reference Laboratory Technical Guidance Document to determine the influence of selected LAB strains in shelf-life studies on L. monocytogenes in RTE foods (such as “Torta del Casar” cheese) where no growth or the growth probability of this pathogen is ≤ 10 % (Beaufort et al., 2014). This paragraph has been included in the revised version of the manuscript.

2.4. Physicochemical analysis – If there are available, some other chemical parameters like fat, protein and proteolysis indexes should be included as they are more important for cheese quality than colour, for instance.

Response:

Other parameters like fat and nitrogen fraction were not determined since the purpose of this work was to evaluate the effect of the addition of selected LAB strains on the evolution of the volatile compounds throughout the ripening process and the final sensory traits. However, to facilitate information to the readers, the protein and lipid content of this type of cheese has been included in the introduction section of the revised manuscript.

Lines 226 – the authors mentioned that the decrease of Enterobacteriaceae throughout the ripening is probably due to antimicrobial effect of the dominant LAB population. This was observed in all batches including the control. This mean that pH, water content and aw decreases has no effect at all? Please discuss this.

Response:

We agree to the Reviewer 4 since a significant (p ≤ 0.05) decrease of Enterobacteriaceae was observed throughout the ripening in all batches, including uninoculated control, probably due to aw decrease and the antimicrobial effect of the dominant LAB population in all these batches. This has been corrected in Results and Discussion section of the revised version of the manuscript.

Round 2

Reviewer 1 Report

I appreciate the efforts of the authors to improve the quality of the manuscrit according to the reviewers' comments, but I still think that the the way of recording homogeneous groups requires improvement by the authors. Please write groups not as a,b but should be ab without comma. Revise the entire manuscript.

Author Response

Reviewer 1.

I appreciate the efforts of the authors to improve the quality of the manuscrit according to the reviewers' comments, but I still think that the way of recording homogeneous groups requires improvement by the authors. Please write groups not as a,b but should be ab without comma. Revise the entire manuscript.

Response:

We thank Reviewer 1 for the comments on the manuscript after the revision. As indicated by the Reviewer 1, the manuscript has been again corrected and revised.

Reviewer 2 Report

Dear authors,

After a detailed review of your manuscript, I find that There are still many points that I think you should provide more explanatory answers.
about:
1. The design of the experimental process (e.g. mode of incorporation of the inoculum ... )
2. Enrichment with more literature mainly on the biochemical and technological properties of the 2 protective bacterial cultures (Lc. casei 116 and Lco. garvieae 151)
3. Determining other categories of cheese microflora and the control of the presence of pathogenic microorganisms Staph. aureus, L. monocytogenes would be necessary to be carried out.
4. Determination of additional necessary parameters, such as salt content, fat content, as well as nitrogen fraction,..etc..
My general impression is that the manuscript in its current state is weak, with substantial limitations.

Author Response

Dear authors,

After a detailed review of your manuscript, I find that There are still many points that I think you should provide more explanatory answers. about:

  1. The design of the experimental process (e.g. mode of incorporation of the inoculum ... )

Response:

In the design of the experimental process the inoculation of the selected LAB strains assayed in the present work was done in the first day of processing of the cheese in the pilot plant of the Faculty of Veterinary Science to avoid contamination in the industry. For this the inoculation was done in the center of the cheese (in a cube of 16 cm2 of surface and 6 cm deep). For the microbiological analysis, the entire cube was used in all sampling times. This was included in the revised version of the manuscript.

  1. Enrichment with more literature mainly on the biochemical and technological properties of the 2 protective bacterial cultures (Lc. casei 116 and Lco. garvieae 151)

Response:

More information about the 2 protective bacterial L. casei and Lc. garvieae was included in the Introduction section of the revised manuscript.

  1. Determining other categories of cheese microflora and the control of the presence of pathogenic microorganisms Staph. aureus, L. monocytogenes would be necessary to be carried out.

Response:

The purpose of the present work was to evaluate the the effect of the addition of selected LAB strains (L. casei and Lc. garvieae) isolated from raw ewe´s milk cheeses on the evolution of the volatile compounds throughout the ripening process and the final sensory traits of “Torta del Casar” cheese as it was indicated in the Introduction section of the revised manuscript. The effect of these strains on control of the presence of pathogenic microorganisms such as L. monocytogenes during processing was already demonstrated in previous work (Martín et al. 2022), as it has been indicted in the revised version of the manuscript.

Martín et al. 2022. Selection and characterization of lactic acid bacteria from traditional ripened foods with activity against Listeria monocytogenes. LWT - Food Sci. Technol., 163, 113579, doi: 10.1016/j.lwt.2022.113579.

Thus, in our opinion determining presence of pathogenic microorganisms is out of purpose of the present work.

  1. Determination of additional necessary parameters, such as salt content, fat content, as well as nitrogen fraction,..etc..

Response:

In the present work were determined those parameters that could influence the growth of the inoculated LAB strains, such as the moisture content, aw and pH. Although salt content was not determined it was aw that it is greatly influenced by salt content. Other parameters like fat and nitrogen fraction were not determined since the purpose of this work was to evaluate the effect of the addition of selected LAB strains on the evolution of the volatile compounds throughout the ripening process and the final sensory traits. However, to facilitate information to the readers, the protein and lipid content of this type of cheese has been included in the introduction section of the revised manuscript.

My general impression is that the manuscript in its current state is weak, with substantial limitations.

Response:

We think that the revised manuscript has been greatly improved with the modifications and suggestions of two revision of the reviewers that have been included in the revised version of the manuscript. In our opinion, the revised version of the manuscript offers interesting information to the reader of Foods and could be now suitable for publishing in this Journal.

Reviewer 3 Report

In its revised form, the manuscript may be submitted for further editorial work

Author Response

In its revised form, the manuscript may be submitted for further editorial work.

Response:

We thank Reviewer 3 for the positive comments on the manuscript.

Reviewer 4 Report

The authors clarify most of the questions. After this review, there are only two main remarks:

Suggestions:

1. Introduction - The introduction must be revised according to the above mentioned.

Authors added “. However, to date, the use of protective cultures in “Torta del Casar” cheese has not been reported in the literature”, but is adding protective cultures currently allowed in PDO Torta del Casar? Please include the answer in the introduction and eventally discuss it.

2.2. Preparation of cheese – It is unclear how many times (how many replicates) the experimental design was repeated. It seems that there is only one trial. If so, it is not adequate for the discussion and the generalization of the results. The trial protocol should be repeated at least three times.

As I see from the description, the experiment consists on only one trial, envolving 3 batches – control (Batch C), inoculated Lc (batch Lc) and inoculated Lg (batch Lg), which are compared on a basis of analysis of five cheeses for batch and each ripening time. Was this experimental design repeated twice or three times? Under the current conditions, the variations discussed only concern the sampling repetitions for each ripening time and not the repetition of cheese making, which would, in my view, give a more robust understanding for the intended objectives.

Author Response

The authors clarify most of the questions. After this review, there are only two main remarks:

Suggestions:

  1. Introduction - The introduction must be revised according to the above mentioned.

Authors added “. However, to date, the use of protective cultures in “Torta del Casar” cheese has not been reported in the literature”, but is adding protective cultures currently allowed in PDO Torta del Casar? Please include the answer in the introduction and eventally discuss it.

Response:

The use of protective cultures is not yet regulated in the PDO “Torta del Casar” cheese (Regulation [CE] 1491/2003 of the European Commission, despite its advantages. We hope that in the next future the use of protective culture can be regulated in this PDO and this work can contribute to this.

The introduction of the revised manuscript has been corrected to include the above suggestion.

Preparation of cheese – It is unclear how many times (how many replicates) the experimental design was repeated. It seems that there is only one trial. If so, it is not adequate for the discussion and the generalization of the results. The trial protocol should be repeated at least three times.

As I see from the description, the experiment consists on only one trial, envolving 3 batches – control (Batch C), inoculated Lc (batch Lc) and inoculated Lg (batch Lg), which are compared on a basis of analysis of five cheeses for batch and each ripening time. Was this experimental design repeated twice or three times? Under the current conditions, the variations discussed only concern the sampling repetitions for each ripening time and not the repetition of cheese making, which would, in my view, give a more robust understanding for the intended objectives.

 Response:

Although the experiment consists on only one trial of ripening, involving 3 batches: control (Batch C), inoculated Lc (batch Lc) and inoculated Lg (batch Lg), and five cheeses analysed in each batch and ripening time, the cheeses used in this work were make in a cheese factory from three different cheese making processed at different moment in day and using raw milk from at least five different farms. This has been included in the new revised version of the manuscript.
